# Effects of HMGB1 on Tricellular Tight Junctions via TGF-β Signaling in Human Nasal Epithelial Cells

**DOI:** 10.3390/ijms22168390

**Published:** 2021-08-04

**Authors:** Kizuku Ohwada, Takumi Konno, Takayuki Kohno, Masaya Nakano, Tsuyoshi Ohkuni, Ryo Miyata, Takuya Kakuki, Masuo Kondoh, Kenichi Takano, Takashi Kojima

**Affiliations:** 1Department of Cell Science, Research Institute for Frontier Medicine, Sapporo Medical University School of Medicine, Sapporo 060-8556, Japan; kizuku020@gmail.com (K.O.); t.konno1225@gmail.com (T.K.); kohno@sapmed.ac.jp (T.K.); masayanakano0606@gmail.com (M.N.); 2Department of Otolaryngology, Sapporo Medical University School of Medicine, Sapporo 060-8556, Japan; tu4shioo92@nifty.com (T.O.); ryomiyata1103@yahoo.co.jp (R.M.); kurokuma1029@yahoo.co.jp (T.K.); kent@sapmed.ac.jp (K.T.); 3Drug Discovery Center, Graduate School of Pharmaceutical Sciences, Osaka University, Suita 565-0871, Japan; masuo@phs.osaka-u.ac.jp

**Keywords:** human nasal epithelial cells, hTERT, 2.5D matrigel culture, tight junctions, angulin-1/LSR, HMGB1, OSM, p63, TGF-β type I receptor inhibitor

## Abstract

The airway epithelium of the human nasal mucosa acts as a physical barrier that protects against inhaled substances and pathogens via bicellular and tricellular tight junctions (bTJs and tTJs) including claudins, angulin-1/LSR and tricellulin. High mobility group box-1 (HMGB1) increased by TGF-β1 is involved in the induction of nasal inflammation and injury in patients with allergic rhinitis, chronic rhinosinusitis, and eosinophilic chronic rhinosinusitis. However, the detailed mechanisms by which this occurs remain unknown. In the present study, to investigate how HMGB1 affects the barrier of normal human nasal epithelial cells, 2D and 2.5D Matrigel culture of primary cultured human nasal epithelial cells were pretreated with TGF-β type I receptor kinase inhibitor EW-7197 before treatment with HMGB1. Knockdown of angulin-1/LSR downregulated the epithelial barrier. Treatment with EW-7197 decreased angulin-1/LSR and concentrated the expression at tTJs from bTJs and increased the epithelial barrier. Treatment with a binder to angulin-1/LSR angubindin-1 decreased angulin-1/LSR and the epithelial barrier. Treatment with HMGB1 decreased angulin-1/LSR and the epithelial barrier. In 2.5D Matrigel culture, treatment with HMGB1 induced permeability of FITC-dextran (FD-4) into the lumen. Pretreatment with EW-7197 prevented the effects of HMGB1. HMGB1 disrupted the angulin-1/LSR-dependent epithelial permeability barriers of HNECs via TGF-β signaling in HNECs.

## 1. Introduction

The airway epithelium of the human nasal mucosa interacts with various environmental agents and acts as a physical barrier that protects against inhaled substances and pathogens via tight junctions (TJs) [1,2]. Nasal epithelial barrier dysfunction contributes to various nasal diseases, including allergies [3,4,5,6,7]. TJs are intercellular junctions adjacent to the apical ends of paracellular spaces and have fence, barrier and signaling functions [8]. TJ proteins are closely involved in cancer, innate immunity, and infectious diseases [8]. TJs are composed of claiudins (CLDNs), occludins (OCLN), JAMs and scaffold proteins such as ZO [9]. Angulin-1/lipolysis-stimulated lipoprotein receptor (LSR) and tricellulin (TRIC) seal the extracellular spaces where three epithelial cells come into contact and have various functions [10]. The family of angulins consists of angulin-1/LSR, angulin-2, (immunoglobulin-like domain-containing receptor 1 [ILDR1]) and angulin-3 (ILDR2) [11]. LSR recruits TRIC to tTJs, and both proteins are required for the full barrier function of epithelial cellular sheets [11]. Downregulation of LSR influences not only the barrier function but also various kinds of signaling [12,13,14]. In the human nasal epithelium, OCLN, JAM-A, ZO-1, ZO-2, CLDN-1, -4, -7, -8, -12, -13, -14 at bicellular TJs (bTj) and TRIC and angulin-1/LSR at tricellular TJs (tTJ) are detected on the surfaces of cells [1,2].

Angubindin-1 is a novel binder to angulin-1 and -3 that was developed from *Clostridium perfringens* iota-toxin, which is a binary toxin comprised of an enzymatic component (Ia) and a receptor-binding component (Ib) [15,16]. A fragment of the domain corresponding to amino acids 421-664 (angubindin-1) modulates the tTJ without causing cytotoxicity, increases intestinal absorption of large molecules and opens the blood-brain barrier via the paracellular route by disrupting tricellulin recruitment [16,17,18]. Angubindin-1 decreases the epithelial barrier and the expression of angulin-1/LSR and increases cell invasion in human pancreatic cancer cell line HPAC [19]. Angubindin-1 reversibly regulates the epithelial barrier and cell migration at tricellular contacts via JNK/cofilin/actin cytoskeleton dynamics [12,16].

High mobility group box-1 (HMGB1) is a non-histone DNA-binding nuclear protein that plays a role in stabilization of DNA and gene transcription [20,21]. HMGB1 is also known as a danger associated molecular pattern (DAMP) and it is released from the nucleus to the extracellular milieu in particular conditions such as autoimmunity, sepsis, and hypoxia [22]. Extracellular HMGB1 binds Toll-like receptors (TLRs) and the receptor for advanced glycation endproducts (RAGE) and induces inflammatory signals [22]. HMGB1 is involved in the induction of nasal diseases in patients with allergic rhinitis (AR), chronic rhinosinusitis (CRS), and eosinophilic chronic rhinosinusitis (ECRS) [23,24,25]. It induces hyperpermeability of human nasal epithelial cells by downregulation of ZO-1, OCLN and CLDN-1 via hypoxia [26]. Serum HMGB1 is increased in patient with severe COVID-19 [27].

TGF-β1 induces epithelial permeability in respiratory epithelial cells [28,29]. HMGB1 expression is increased by TGF-β1 and knockdown of HMGB1 reverses TGF-β1-induced epithelial–mesenchymal transition (EMT) in human alveolar epithelial cell line A549 and normal human bronchial epithelial cell line BEAS-2B [30]. HMGB1 enhances epithelial permeability via p63/TGF-β signaling in lung and terminal bronchial epithelial cells [31]. The disruption of the epithelial barrier induced by HMGB1 and inflammatory cytokines contributes to TGF-β/EGF signaling in Caco-2 cells [32]. 

EW-7197 is a transforming growth factor-β type I receptor kinase inhibitor with potential anti-inflammatory and antifibrotic properties [33]. In airway cell line Calu-3, HMGB1 downregulates angulin-1/LSR expression and the epithelial barrier, and upregulates claudin-2 expression, AMPK activity and mitochondrial respiration, and EW-7197 prevents all the effects of HMGB1 [31]. In normal human lung epithelial cells, EW-7197 prevents hyperpermeability and the changes of all tight junction molecules induced by HMGB1 [31]. However, it is not yet known whether EW-7197 prevents the epithelial barrier disruption triggered by HMGB1 in nasal inflammation and injury.

Transcriptional factor p63, which is a member of the p53 family and has two distinct isoforms, TAp63 and ΔNp63, plays an important role in the proliferation and differentiation of various epithelial basal cells [34]. Whereas TAp63 isoform is capable of transactivating p53 target genes and inducing apoptosis, ΔNp63 isoform acts in a dominant-negative fashion to counteract the transactivation-competent isoforms of p63 and p53. p63 negatively regulates the epithelial tight junctional barrier of the nasal epithelium [35] and is upregulated in the epithelium of CRS and nasal polyps [35]. In normal human lung epithelial cells, knockdown of p63 increases angulin-1/LSR and CLDN-4 and prevents the hyperpermeability induced by HMGB1 as well as pretreatment with EW-7197 [31]. 

Human telomerase reverse transcriptase (hTERT)-transfected human nasal epithelial cells (hTERT-HNECs) can be used as a stable model for studying regulation of the nasal epithelial response [36,37,38]. Furthermore, two and one-half dimensional (2.5D) culture in which the cells are plated with additional 10% Matrigel in the medium on top of 100% Matrigel, induces the cells to form a more physiological tissue architecture than 2D culture in a stromal matrix such as collagen I and the cells remain accessible for molecular analysis. In the present study, to investigate how HMGB1 affects the barrier of normal human nasal epithelial cells, monolayer (2D culture) and spheroid (2.5D Matrigel culture) hTERT-HNECs were pretreated with TGF-β type I receptor kinase inhibitor EW-7197 or pretransfected with siRNA-p63 before treatment with HMGB1 and the changes of the epithelial permeability barriers and bTJ and tTJ molecules were examined. This was done to promote the possible development of a therapy for nasal inflammation and injury in patients with allergic rhinitis, chronic rhinosinusitis, and eosinophilic chronic rhinosinusitis.

## 2. Results

### 2.1. Knockdown of Angulin-1/LSR Downregulates Claudin-7 and Epithelial Barrier Function and Upregulates of CLDN-1 and -4 in Primary Cultured Human Nasal Epithelial Cells (HNECs)

To investigate the roles of angulin-1/LSR in the epithelial barrier of human nasal epithelia, we used primary cultured human nasal epithelial cells (HNECs) isolated from nasal mucosal tissues of patient with hypertrophic rhinitis or chronic sinusitis who underwent inferior turbinectomy. Immunocytochemistry showed expression of tTJ molecules angulin-1/LSR and TRIC at both the areas of tricellular tight junctions (tTJs) and bicellular tight junctions (bTJs) in HNECs cultured with 10% FBS (Figure 1A). In RT-PCR analysis, mRNAs of angulin-1/LSR and TRIC were detected as well as mRNAs of bTJ molecules CLDN-1, -4, and -7 in HNECs cultured with or without 10% FBS (Figure 1B).

For knockdown of angulin-1/LSR expression in HNECs, HNECs were transfected with siRNA-LSR and scrambled siRNA as a control for 48 h. Western blotting revealed that knockdown of angulin-1/LSR increased CLDN-1 and -4 and decreased CLDN-7 in HNECs without 10% FBS, while no change of TRIC expression was observed (Figure 1C). In immunocytochemistry of the knockdown cells, angulin-1/LSR expression disappeared at the membranes and expression of CLDN-1 and -4 was increased (Figure 1D). Knockdown of angulin-1/LSR significantly decreased transepithelial electrical resistance (TEER) values used to indicate the epithelial barrier, compared with the scrambled siRNA controls (Figure 1E).

### 2.2. TGF-β Receptor Type 1 Inhibitor EW-7197 Concentrates Expression of Angulin-1/LSR and TRIC at tTJs from bTJs and Increases Epithelial Barrier in HNECs

The TGF-β receptor type 1 inhibitor EW-7197 induces the epithelial barrier with an increase of anguin-1/LSR expression at the membranes in primary cultured normal human lung epithelial cells [31]. To investigate the regulation of angulin-1/LSR expression in human nasal epithelia, HNECs were treated with 10 μM EW-7197 for 24 h. Phase-contrast imaging revealed that cells treated with EW-7197 changed to an epitheloid-like shape (Figure 2A). Western blot analysis showed that treatment with EW-7197 decreased expression of angulin-1/LSR, TRIC, and p63 and increased expression of CLDN-1 and -4 (Figure 2B). Treatment with EW-7197 enhanced the epithelial barrier function compared to the control (Figure 2C). In immunocytochemistry, the expression of angulin-1/LSR and TRIC was concentrated at the areas of tTJs from bTJs by treatment with EW-7197, while no change of OCLN and CLDN-4 was observed at the membranes (Figure 2D).

To investigate whether angulin-1/LSR is regulated via TGF-β signaling, HNECs were pretreated with 10 μM EW-7197 before treatment with 100 ng/mL TGF-β1 (Figure 3A). Western blot analysis showed that treatment with TGF-β1 increased expression of angulin-1/LSR, CLDN-4, and phospho-MAPK, and decreased expression of CLDN-1 and -7 compared to the control, while expression of TRIC, HMGB1, and phosphor-AMPK was not changed (Figure 3B). Pretreatment with EW-7197 prevented all the changes induced by treatment with TGF-β1 (Figure 3B).

### 2.3. A Binder to Angulin-1/LSR Angubindin-1 Decreases Anglin-1/LSR and Epithelial Barrier Function in HNECs

A binder to angulin-1/LSR angubindin-1 decreases the epithelial barrier function at tTJs in various types of cells [12,16,18,19]. To investigate the effects of angubindin-1 on the epithelial barrier of human nasal epithelia, HNECs were treated with 2.5 μg/mL angubindin-1 for 24 h. Western blot analysis showed that angubindin-1 decreased angulin-1/LSR expression, and increased expression of CLDN-4, -7, p63, and phospho-AMPK (pAMPK) (Figure 4A). Immunocytochemical analysis showed that treatment with angubindin-1 decreased angulin-1/LSR but not TRIC at the tTJs (Figure 4B). TEER values were decreased in HNECs treated with angubindin-1 (Figure 4C). Treatment with the TGF-β receptor type 1 inhibitor EW-7197 prevented upregulation of CLDN-4 and pAMPK (Figure 4D).

### 2.4. HMGB1 Decreases Angulin-1/LSR Expression and Epithelial Barrier Function in HNECs

HMGB1 closely contributes to chronic rhinosinusitis with or without nasal polyps [39,40]. Furthermore, HMGB1 disrupts the airway epithelial barrier of human bronchial and lung epithelia via angulin-1/LSR [31,41]. To investigate whether HMGB1 affects the epithelial barrier in human nasal epithelia, HNECs were treated with 100 ng/mL HMGB1 for 24 h. Treatment with HMGB1 decreased expression of angulin-1/LSR, CLDN-4, and p63 in Western blotting (Figure 5A). TEER values were significantly decreased by treatment with HMGB1 compared to the control (Figure 5B). Immunocytochemical analysis showed that HMGB1 downregulated expression of angulin-1/LSR and OCLN at the membranes and OCLN-positive vesicles were scattered in the cytoplasm (Figure 5C).

### 2.5. TGF-β Receptor Type 1 Inhibitor EW-7197 and Knockdown of p63 Suppressed the Effects of HMGB1 in HNECs

The TGF-β receptor type 1 inhibitor EW-7197 prevents the disruption of the epithelial barrier and induction of epithelial permeability by HMGB1 in airway epithelial cells [31,41]. To investigate whether EW-7197 prevents the effects of HMGB1 on the epithelial barrier and tight junction molecules in human nasal epithelia, HNECs were pretreated with 10 μM EW-7197 before treatment with 100 ng/mL HMGB1 for 24 h. In Western blotting, treatment with EW-7197 prevented the downregulation of angulin-1/LSR, TRIC, CLDN-4, and p63 induced by treatment with HMGB1 (Figure 6A). In 2.5D culture of HNECs, treatment with HMGB1 induced the permeability of FD-4 into the lumen and pretreatment with EW-7197 prevented the hyperpermeability of FD-4 into the lumen induced by HMGB1 (Figure 6B).

### 2.6. Knockdown of p63 Prevents Downregulation of Angulin-1/LSR Induced by HMGB1 in HNECs

In HNECs, knockdown of p63 by siRNAs of TAp63 and ΔNp63 induces CLDN-1 and -4 with Sp1 activity and enhances the barrier and fence functions [35]. In the present study, knockdown of p63 by siRNA of TAp63 induced angulin-1/LSR and OCLN at the membranes in HNECs cultured without 10% FBS (Figure 6C). Treatment with HMGB1 decreased expression of angulin-1/LSR and OCLN at the membranes and OCLN-positive vesicles were scattered in the cytoplasm (Figure 6C). Western blot analysis showed that knockdown of p63 increased expression of angulin-1/LSR and CLDN-4, and decreased TRIC expression (Figure 6D). Treatment with HMGB1 enhanced the upregulation of angulin-1/LSR and downregulation of p63 induced by knockdown of p63 (Figure 6D). 

### 2.7. Knockdown of HMGB1 Induces Angulin-1/LSR and Enhances Epithelial Barrier Function in HNECs

To investigate the effects of knockdown of HMGB1 on the epithelial barrier and TJ molecules in human nasal epithelia, HNECs were transfected with siRNA of HMGB1 and scrambled siRNA as a control for 48 h. Western blotting revealed that knockdown of HMGB1 increased expression of angulin-1/LSR and TRIC, while no change of CLDN-1, -4, and -7 was observed (Figure 7A). In immunocytochemical analysis of HMGB1-knockdown cells, expression of angulin-1/LSR and OCLN was observed as clear lines at the membranes compared to the control. OCLN expression was not changed (Figure 7B). Knockdown of HMGB1 significantly increased the epithelial barrier function measured as TEER values (Figure 7C).

### 2.8. Effects of Oncostatin M on Epithelial Barrier and TJ Molecules in HNECs

Oncostatin M (OSM) is also an inflammatory mediator associated with chronic rhinosinusitis with nasal polyps and it promotes epithelial barrier dysfunction [42]. To investigate the effect of OSM on the epithelial barrier and TJ molecules in human nasal epithelia, HNECs were treated with 100 ng/mL OSM with or without 10 μM EW-7197 for 24 h (Figure 8A). Treatment with OSM and/or EW-7197 increased the TEER values (Figure 8B). In immunocytochemistry of treatment with OSM with or without EW-7197, angulin-1/LSR and OCLN were observed at both the areas of tTJs and bTJs, like those of the control, whereas in treatment with EW-7197, angulin-1/LSR was strongly expressed at the tTJs (Figure 8C). Western blot analysis showed that treatment with OSM decreased CLDN-1 expression, whereas pretreatment with EW-7197 prevented the downregulation of CLDN-1 induced by OSM (Figure 8D). Treatment with EW-7197 with or without OSM increased CLDN-4, -7 and pAMPK, whereas treatment with OSM and EW-7197 decreased angulin-1/LSR (Figure 8D).

### 2.9. Changes of COVID-19 Infection-Related Genes Induced by Treatment with HMGB1 and OSM in HNECs

Serum HMGB1 in patient with severe COVID-19 is increased and exogenous HMGB1 induces the expression of SARS-CoV-2 entry receptor ACE2 in alveolar epithelial cells [27]. To investigate the effects of HMGB1 and OSM on COVID-19 infection-related genes of HNECs, we performed GeneChip analysis of HNECs treated with 100 ng/mL HMGB1 and 100 ng/mL OSM and selected gene probes (Table 1). In HNECs treated with HMGB1 and OSM, upregulation of the COVID-19-related genes TMPRSS6, furin, and cathepsin L (CTSL) was observed compared to the control, while no change of ACE2 or TMPRSS2 was observed.

## 3. Discussion

In the present study by using normal HNECs, knockdown of angulin-1/LSR decreased the epithelial barrier function. A binder to angulin-1/LSR angubindin-1 decreased angulin-1/LSR and the epithelial barrier. The TGF-β type I receptor kinase inhibitor EW-7197 decreased angulin-1/LSR and concentrated the expression at tTJs from bTJs and increased the epithelial barrier. HMGB1 decreased angulin-1/LSR and epithelial barrier and increased epithelial permeability in 2D and 2.5D cultures. EW-7197 prevented the disruption of the epithelial barrier and hyperpermeability induced by HMGB1 with the changes of TJ molecules, including angulin-1/LSR, in HNECs. 

Angulin-1/LSR is localized at tTJs and has an important role in epithelial barrier function [43]. Downregulation of angulin-1/LSR induces epithelial barrier dysfunction in various cells [12,19]. Angubindin-1 is a novel binder to angulin-1/LSR and -3 and that causes the removal of angulin-1 and TRIC from the tTJ which enhanced the permeation of macromolecular solutes [16]. Angubindin-1 reversibly decreases angulin-1/LSR and blood-brain barrier integrity [17]. In the present study of HNECs, knockdown of angulin-1/LSR led to a decrease of epithelial barrier function with upregulation of CLDN-1 and -4 and downregulation of CLDN-7 in HNECs. Angubindin-1 decreased angulin-1/LSR and the epithelial barrier and increased CLDN-4 and -7. These findings show that angulin-1/LSR at tTJs plays a crucial role in the epithelial barrier with other tight junction molecules at bTJs in HNECs.

HMGB1 is one of the damage-associated molecular patterns (DAMPs) and is also a proinflammatory mediator that belongs to the alarmin family [20]. It is abundantly and widely expressed in a variety of cell nuclei and plays a role in gene transcription in various human diseases, including autoimmune diseases, inflammatory diseases and cancers [21]. HMGB1 promotes the induction of inflammatory cytokines in the pathogenesis of various inflammatory diseases [23]. Patients with severe symptoms of chronic sinusitis have high HMGB1 serum levels [24]. It increases mesenchymal markers and decreases epithelial markers in HNECs and contributes to nasal polyps [44]. HMGB1 induces hyperpermeability of HNECs by downregulation of ZO-1, OCLN, and CLDN-1 via hypoxia [26]. In normal human lung epithelial cells, treatment with HMGB1 decreases the epithelial barrier function with downregulation of angulin-1/LSR, TRIC, and CLDN-1, -4, -7 and upregulation of CLDN-2 [31]. HMGB1 expression is increased by TGF-β1 and knockdown of HMGB1 reverses the TGF-β1-induced epithelial–mesenchymal transition (EMT) in human alveolar epithelial cell line A549 and normal human bronchial epithelial cell line BEAS-2B [30]. It also induces hyperpermeability via TGF-β/EGF signaling in 2.5D culture of Caco-2 cells [32]. In the present study, HMGB1 downregulated angulin-1/LSR and CLDN-4, and disrupted the epithelial barrier function in HNECs in 2D culture. In 2.5D Matrigel culture of HNECs, HMGB1 induced hyperpermeability of FD-4 into the lumen. In 2D culture, knockdown of HMGB1 strengthened the epithelial barrier via upregulation of angulin-1/LSR and TRIC. These results indicated that exogenous HMGB1 disrupted the epithelial barrier and induced epithelial permeability of HNECs via downregulation of TJ molecules, including angulin-1/LSR, whereas endogenous HMGB1 slightly decreased the epithelial barrier function by downregulation of tTJ molecules in HNECs in vitro.

EW-7197 is known as a transforming growth factor-β type I receptor kinase inhibitor [45]. It prevents changes of the distribution and the barrier function of angulin-1/LSR by TGF-β1 [19]. In airway cell line Calu-3, HMGB1 downregulates angulin-1/LSR expression and the epithelial barrier, and upregulates claudin-2 expression, AMPK activity and mitochondrial respiration, and EW-7197 prevents all these effects of HMGB1 [41]. In normal human lung epithelial cells, pretreatment with EW-7197 prevents the downregulation of tight junction molecules induced by HMGB1 in 2D culture and the hyperpermeability of FD-4 into the lumen induced by HMGB1 in 2.5D culture [31]. EW-7197 also prevents the hyperpermeability induced by HMGB1 and inflammatory cytokines in 2.5D cultured of Caco-2 cells [32]. In the present study of HNECs, treatment with EW-7197 alone decreased angulin-1/LSR, and concentrated its expression at tTJs from bTJs and increased the epithelial barrier function. It prevented the downregulation of angulin-1/LSR, TRIC, and CLDN-4 induced by HMGB1 in 2D culture and the disruption of epithelial barrier and hyperpermeability induced by HMGB1 in 2D culture and 2.5D culture. These results suggested that, in HNECs, HMGB1 disrupted the angulin-1/LSR-dependent epithelial permeability barriers of HNECs via TGF-β signaling and that EW-7197 might have potential for use in therapy for the disruption of the epithelial permeability barriers induced by HMGB1.

The transcription factor p63, a component of the p53 family, plays important roles in the development, homeostasis, and regeneration of epithelial tissues [34,44]. p63 is upregulated in the epithelium of CRS and nasal polyps [46,47]. It negatively regulates the epithelial tight junctional barrier of the nasal epithelium [47]. In normal human lung epithelial cells, knockdown of p63 increases angulin-1/LSR and CLDN-4 and prevents the hyperpermeability induced by HMGB1 [31]. In the present study of HNECs, treatment with EW-7197 decreased p63 and increased the epithelial barrier. Treatment with a binder to angulin-1/LSR angubindin-1 decreased the epithelial barrier and increased p63. Treatment with HMGB1 decreased p63 and the epithelial barrier. Knockdown of HMGB1 induced p63 and the epithelial barrier. Knockdown of p63 increased expression of angulin-1/LSR and CLDN-4, decreased TRIC expression and induced angulin-1/LSR and OCLN at the membranes of HNECs. Treatment with HMGB1 further enhanced the downregulation of p63 induced by knockdown of p63. These findings indicated that, in HNECs, p63 controlled via TGF-β signaling regulated the epithelial tight junctional barrier, including angulin-1/LSR expression, and that HMGB1 in part induced the epithelial barrier disruption via p63. 

Oncostatin M (OSM) is one of the interleukin 6 family cytokines secreted by T lymphocytes, neutrophils, and macrophages [48]. It is an inflammatory mediator associated with chronic rhinosinusitis with nasal polyps and it promotes epithelial barrier dysfunction [42,49,50]. In HNECs, treatment with OSM increased the epithelial barrier and decreased CLDN-1 expression, whereas pretreatment with EW-7197 prevented the downregulation of CLDN-1 induced by OSM. In the present study, OSM did not promote epithelial barrier dysfunction in HNECs. 

Serum HMGB1 is increased in patients with severe COVID-19 and exogenous HMGB1 induces the expression of SARS-CoV-2 entry receptor ACE2 in alveolar epithelial cells [27]. In HNECs treated with HMGB1 and OSM, upregulation of the COVID-19-related genes TMPRSS6, furin, and cathepsin L (CTSL) is observed. Cysteine protease CTSL expression is upregulated during chronic inflammation and is involved in processing the COVID-19 spike protein [51]. The protease furin is also involved in mediating SARS-CoV-2 entry [52]. It is possible that HMGB1 induced by COVID-19 infection may increase in HNECs and disrupt the epithelial permeability barriers. Furthermore, it is also thought that the risk of COVID-19 infectivity may also be increased by HMGB1 induced during AR, CRS, and ECRS.

In conclusion, our findings indicated that HMGB1 affected the epithelial permeability barrier and tTJ molecule angulin-1/LSR via TGF-β signaling in normal HNECs. Treatment with TGF-β type I receptor kinase inhibitor EW-7197 prevented the disruption of the epithelial permeability barrier by HMGB1 in HNECs. EW-7197 may have potential for use in therapy for the disruption of the epithelial barrier induced by HMGB1 in patients with AR, CRS, and ECRS. 2.5D Matrigel culture of HNECs is similar to nasal epithelial structures in vivo and might be a useful in vitro model for studying HMGB1 and other nasal diseases.

## 4. Materials and Methods

### 4.1. Antibodies and Reagents

A rabbit polyclonal anti-lipolysis-stimulated lipoprotein receptor (LSR) antibody was obtained from Novus Biologicals (Littleton, CO, USA). A mouse monoclonal anti-LSR antibody was from Abnova (Taipei, Taiwan). A monoclonal anti-CD11c antibody was from DAKO (Carpinteria, CA, USA). Rabbit polyclonal anti-claudin (CLDN)-1, anti-CLDN-4, anti-CLDN-7, anti-tricellulin (TRIC) antibodies and mouse monoclonal anti-occludin (OC-3F10, OCLN), anti-CLDN-1(2H10D10), and anti-CLDN-4 (3E2C1) antibodies were from Zymed Laboratories (San Francisco, CA, USA). Rabbit polyclonal anti-p63 and anti-HMGB1 antibodies were from Abcam (Cambridge, MA, USA). A rabbit polyclonal anti-actin antibody and recombinant human HMGB1 were from Sigma-Aldrich Inc. (St. Louis, MO, USA). Rabbit polyclonal anti-phosphorylated-AMPK was from Cell Signaling Technology (Danvers, MA, USA). A rabbit polyclonal anti-actin antibody was from Sigma-Aldrich Inc. (St. Louis, MO, USA). The TGF-β receptor type 1 inhibitor EW-7197 (N-((4-([1,2,4] triazolo [1,5-a] pyridin-6-yl)-5-(6-methylpyridin-2-yl)-1H-imidazol-2-yl) methyl)-2-fluoroaniline) was from Cayman Chemical (Ann Arbor, MI, USA). Recombinant human TGF-β and oncostatin M were from PeproTech (London, UK). Alexa Fluor 488 (green)-conjugated anti-rabbit IgG, and Alexa Fluor 594 (red)-conjugated anti-mouse IgG antibodies and Axea Fluor 594 (red)-phalloidin were from Molecular Probes, Inc. (Eugene, OR, USA). FITC-dextran (FD-4, MW 4.0 kDa) was from Sigma-Aldrich Co. (St. Louis, MO, USA). The ECL Western blotting system was from GE Healthcare UK, Ltd. (Buckinghamshire, UK).

### 4.2. Preparation of Angubindin-1

Angubindin-1 was a kind gift from the Drug Discovery Center, Graduate School of Pharmaceutical Sciences, Osaka University, Osaka, Japan as part of our joint research [14,15]. The plasmid encoding angubindin-1 (pGEX-Ib421–664) was expressed as a fusion protein with glutathione S-transferase (GST) in *Escherichia coli* strain BL21 (DE3). After growth at 37 °C and induction with isopropyl β-D-thiogalactopyranoside (Nacalai Tesque, Kyoto, Japan) of a 1000-mL culture, cells were harvested, resuspended in buffer F (50 mM Tris-HCl [pH 7.5], 150 mM NaCl, 2 mM CaCl_2_), and then lysed by sonication. The lysates were applied to Glutathione Sepharose 4B beads (GE Healthcare, Buckinghamshire, UK) and purified GST-tagged angubindin-1 was cleaved with thrombin. After removal of the thrombin using Benzamidine Sepharose 4 Fast Flow (GE Healthcare), the solvent for angubindin-1 was changed to phosphate-buffered saline (PBS) by gel filtration using a PD-10 column (GE Healthcare), and the purified proteins were stored at −80 °C until use. The concentration of the purified proteins was determined using a BCA protein assay kit (Thermo Fisher Scientific, Waltham, MA, USA) with bovine serum albumin (BSA) as the standard. Purification of the recombinant proteins was confirmed by sodium dodecyl sulfate–polyacrylamide gel electrophoresis (SDS-PAGE). To investigate the distribution of angubindin-1 in the brain, a DyLight 550 antibody labeling kit (Thermo Fisher Scientific; Waltham, MA, USA) was used to label angubindin-1 with DyLight 550.

### 4.3. GeneChip Analysis

Microarray slides were scanned using a 3D-Gene human Oligo chip 25k. (TORAY, Tokyo, Japan) and microarray images were automatically analyzed using AROS^TM^, version 4.0 (Operon Biotechnologies, Tokyo, Japan).

### 4.4. Immunohistochemical Analysis

Human nasal tissues were obtained from patients with hypertrophic rhinitis or chronic sinusitis who underwent inferior turbinectomy at the Sapporo Hospital of Hokkaido Railway Company. Informed consent was obtained from all patients and this study was approved by the ethics committee of the above institution. The tissues were embedded in paraffin after fixation with 10% formalin in PBS. Briefly, 5-μm-thick sections were dewaxed in xylene, rehydrated in ethanol, and heated with Vision BioSystems Bond Max using ER2 solution (Leica) in an autoclave for antigen retrieval. Endogenous peroxidase was blocked by incubation with 3% hydrogen peroxide in methanol for 10 min. The tissue sections were then washed twice with Tris-buffered saline (TBS) and preblocked with Block Ace for 1 h. After washing with TBS, the sections were incubated with anti-LSR (1:100), anti-TRIC (1:100), and anti-CD11c (1:100) antibodies for 1 h. The sections were then washed three times in TBS and incubated with Vision BioSystems Bond Polymer Refine Detection kit DS9800. After three washes in TBS, a diamino-benzidine tetrahydrochloride working solution was applied. Finally, the sections were counterstained with hematoxylin. 

### 4.5. Cell Culture and Treatments

The cultured HNECs were obtained from patients with hypertrophic rhinitis or chronic sinusitis who underwent inferior turbinectomy at the Sapporo Hospital of Hokkaido Railway Company. Informed consent was obtained from all patients and this study was approved by the ethics committee of the above institution. The methods for primary culture of human nasal epithelial cells were as reported previously [34]. Some primary cultured HNECs were transfected with the catalytic component of telomerase, the human catalytic subunit of the telomerase reverse transcriptase (hTERT) gene, as described previously [34]. The cells were plated on 35-mm or 60-mm culture dishes (Corning Glass Works, Corning, NY, USA), which were coated with rat tail collagen (500 μg of dried tendon/mL 0.1% acetic acid). They were then cultured in serum-free bronchial epithelial cell basal medium (BEBM, Lonza Walkersville, Inc.; Walkersville, MD, USA) supplemented with bovine pituitary extract (1% *v/v*), 5 μg/mL insulin, 0.5 μg/mL hydrocortisone, 50 μg/mL gentamycin, 50 μg/mL amphotericin B, 0.1 ng/mL retinoic acid, 10 μg/mL transferrin, 6.5 μg/mL triiodothyronine, 0.5 μg/mL epinephrine, 0.5 ng/mL epidermal growth factor (Lonza Walkersville, Inc.; Walkersville, MD, USA), 100 U/mL penicillin, and 100 μg/mL streptomycin (Sigma-Aldrich; St. Louis, MO, USA) and incubated in a humidified, 5% CO_2_:95% air incubator at 37 °C. In this experiment, second and third passaged cells that maintained cellular function were used. Some HNECs were treated with 10 μM EW-7197, 2.5 μg/mL angubindin-1, 100 ng/mL TGF-β1, 100 μM HMGB1 and 100 ng/mL OSM.

### 4.6. 2.5-Dimensional (2.5D) Matrigel Culture

Thirty-five Millimeter culture dishes or 35-mm culture glass-coated dishes were coated with 100% Matrigel (30 μL or 15 μL) at 4 °C and incubated at 37 °C for 30 min. HNECs (5 × 10^4^) were plated in BEGM medium with 10% Matrigel and cultured for 4 days in BEGM medium without FBS. Some cells were pretreated with 10 μM EW-7197 before treatment with 100 μM HMGB1. In all experiments, 10 spheroids were examined.

### 4.7. siRNA Experiment

For RNA interference studies, small interfering RNA (siRNA) duplexes targeting the mRNA sequences of human LSR and p63 were purchased from Thermo Fisher Scientific (Waltham, MA, USA). The sequences were as follows: siRNA of LSR (forward sense 5′-CCCACGCAACCCAUCGUCAUCUGGA-3′, reverse sense 5′-UCCAGAUGACGAUGGGUUGCGUGGG-3′), siRNA of p63 (forward sense 5′-CCCACGCAACCCAUCGUCAUCUGGA-3′, reverse sense 5′-UCCAGAUGACGAUGGGUUGCGUGGG-3′), siRNA of HMGB1 (forward sense 5′-CCCACGCAACCCAUCGUCAUCUGGA-3′, reverse sense 5′-UCCAGAUGACGAUGGGUUGCGUGGG-3′). hTERT-HNECs cultured at 24 h after plating were transfected with 100 nM siRNAs using Lipofectamine^TM^ RNAiMAX Reagent (Invitrogen) for 48 h. A scrambled siRNA sequence (BLOCK-iT Alexa Fluor fluorescent, Invitrogen) was employed as control siRNA.

### 4.8. RNA Isolation and Reverse Transcription Polymerase Chain Reaction (RT-PCR) Analysis 

Total RNA was extracted and purified using TRIzol (Invitrogen, Carlsbad, CA, USA). One microgram of total RNA was reverse-transcribed into cDNA using a mixture of oligo (dT) and Superscript II reverse transcriptase according to the manufacturer’s recommendations (Invitrogen). Synthesis of each cDNA was performed in a total volume of 20 μL for 50 min at 42 °C and terminated by incubation for 15 min at 70 °C. PCR was performed in a 20-μL total mixture containing 100 pM primer pairs, 1.0 μL of the 20-μL total RT product, PCR buffer, dNTPs, and Taq DNA polymerase according to the manufacturer’s recommendations (Takara, Kyoto, Japan). Amplifications were for 25–35 cycles depending on the PCR primer pair with cycle times of 15 s at 96 °C, 30 s at 55 °C and 60 s at 72 °C. Final elongation time was 7 min at 72 °C. Seven microliters of the total 20-µL PCR product was analyzed by 1% agarose gel electrophoresis with ethidium bromide staining and standardized using a GeneRuler^TM^ 100 bp DNA ladder (Fermentas, ON, Canada). The PCR primers used to detect LSR, TRIC, CLDN-1, -4, -7 and glucose-3-phosphate dehydrogenase (G3PDH) had the sequences shown in Table 2.

### 4.9. Western Blot Analysis

The hTERT-transfected HNECs were scraped from a 60 mm dish containing 300 μL of buffer (1 mM NaHCO3 and 2 mM phenylmethylsulfonyl fluoride), collected in microcentrifuge tubes, and then sonicated for 10s. The protein concentrations of the samples were determined using a BCA protein assay reagent kit (Pierce Chemical Co.; Rockford, IL, USA). Aliquots of 15 μL of protein/lane for each sample were separated by electrophoresis in 5–20% SDS polyacrylamide gels (Wako, Osaka, Japan), and electrophoretically transferred to a nitrocellulose membrane (Immobilon; Millipore Co.; Bedford, UK). The membrane was saturated for 30 min at room temperature with blocking buffer (25 mM Tris, pH 8.0, 125 mM NaCl, 0.1% Tween 20, and 4% skim milk) and incubated with anti-LSR, anti-TRIC, anti-CLDN-1, anti-CLDN-4, anti-CLDN-7, anti-HMGB1, anti-phospho-p44/p42 MAPK, anti-phospho-AMPK and anti-actin antibodies (1:1000) at room temperature for 1 h. Then it was incubated with HRP-conjugated anti-mouse and anti-rabbit IgG antibodies at room temperature for 1 h. The immunoreactive bands were detected using an ECL Western blotting system. 

### 4.10. Immunocytochemistry

hTERT-transfected HNECs grown in 35mm glass-coated wells (Iwaki, Chiba, Japan), were fixed with cold acetone and ethanol (1:1) at −20 °C for 10 min. After rinsing in PBS, the cells were incubated with anti-LSR, anti-TRIC, anti-CLDN-1, anti-CLDN-4, anti-OCLN, and anti-p63 antibodies (1:100) overnight at 4 °C. Alexa Fluor 488 (green)-conjugated anti-rabbit IgG and Alexa Fluor 592 (red)-conjugated anti-mouse IgG (Invitrogen) were used as secondary antibodies. The specimens were examined and photographed with an Olympus IX 71 inverted microscope (Olympus Co.; Tokyo, Japan) and a confocal laser scanning microscope (LSM510; Carl Zeiss, Jena, Germany). 

### 4.11. Measurement of Transepithelial Electrical Resistance (TEER)

hTERT-transfected HNECs were cultured to confluence in the inner chambers of 12-mm Transwell inserts with 0.4-µm pore-size filters (Corning Life Sciences; Tewksbury, MA, USA). TEER was measured using an EVOM voltameter with an ENDOHM-12 (World Precision Instruments; Sarasota, FL, USA) on a heating plate (Fine; Tokyo, Japan) adjusted to 37 °C. The values were expressed in standard units of ohms per square centimeter and presented as the mean ± S.D. For calculation, the resistance of blank filters was subtracted from that of filters covered with cells.

### 4.12. Fluorescein Isothiocyanate (FITC) Permeability Assay

To assess barrier function, the permeability of fluorescein isothiocyanate (FITC)-dextran (FD-4, MW 4.0 kDa) from the outside into the spheroid lumen was examined by using 2.5D Matrigel culture of HNECs on 35-mm glass-coated dishes. Then HNEC cells in 2.5D Matrigel culture were incubated in the medium with 1% FD-4 at 37 °C for 2 h. Ten spheroids of all experiments were photographed and measured by a confocal laser scanning microscope with imaging software (LSM5 PASCAL; Carl Zeiss, Jena, Germany).

### 4.13. Data Analysis

Signals were quantified using Scion Image Beta 4.02 Win (Scion Co.; Frederick, MA, USA). Each set of results shown is representative of at least three separate experiments. Results are given as means ± SEM. Differences between groups were tested by ANOVA followed by a post hoc test and an unpaired two-tailed Student’s *t*-test and considered to be significant when *p* < 0.05.

## Figures and Tables

**Figure 1 ijms-22-08390-f001:**
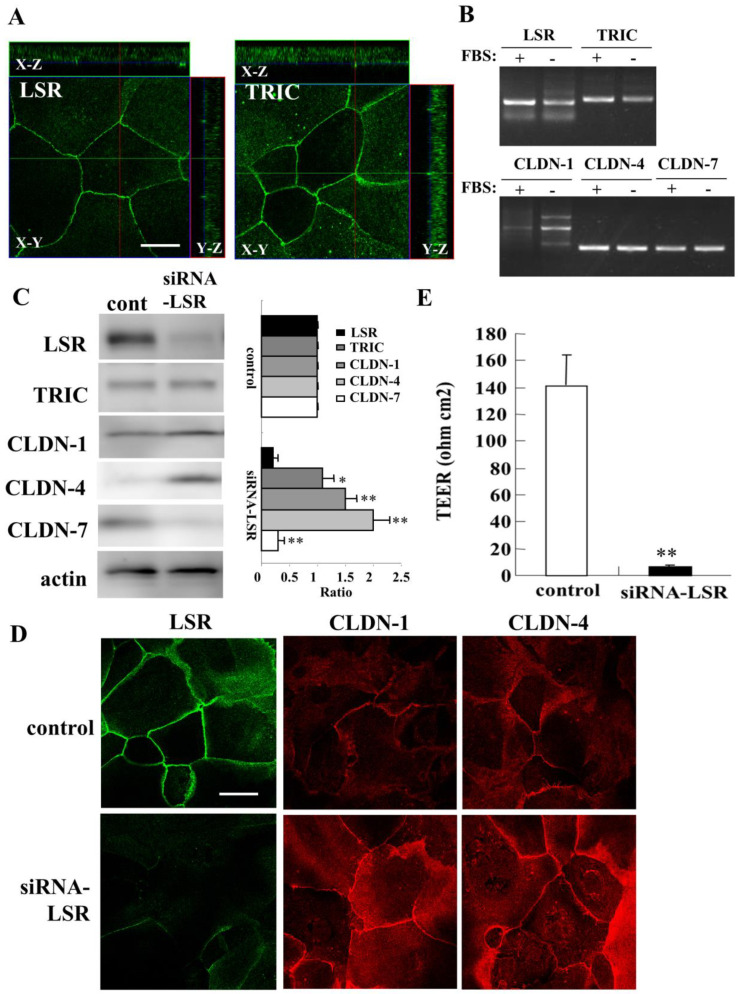
Characterization of expression of tricellular tight junction molecules in HNECs cultured with or without 10% FBS and the effects induced by knockdown of angulin-1/LSR on tight junction molecules and epithelial barrier function. (**A**) Images of immunocytochemical staining for angulin-1/LSR and TRIC in HNECs. Scale bar: 20 µm. (**B**) RT-PCR for angulin-1/LSR, TRIC, CLDN-1, -4, and -7 in HNECs with/without 10% FBS. (**C**) Western blotting for tight junction molecules transferred with siRNA-angulin-1/LSR for 48 h in HNECs without 10% FBS. The corresponding expression levels are shown as bar graphs. * *p* < 0.05, ** *p* < 0.01, vs. control. (**D**) Images of immunocytochemical staining for angulin-1/LSR, CLDN-1 and -4 by knockdown of angulin-1/LSR in HNECs with 10%FBS. Scale bar: 20 μm. (**E**) TEER values representing epithelial barrier function after knockdown of angulin-1/LSR in HNECs with 10%FBS. ** *p* < 0.01, vs. control.

**Figure 2 ijms-22-08390-f002:**
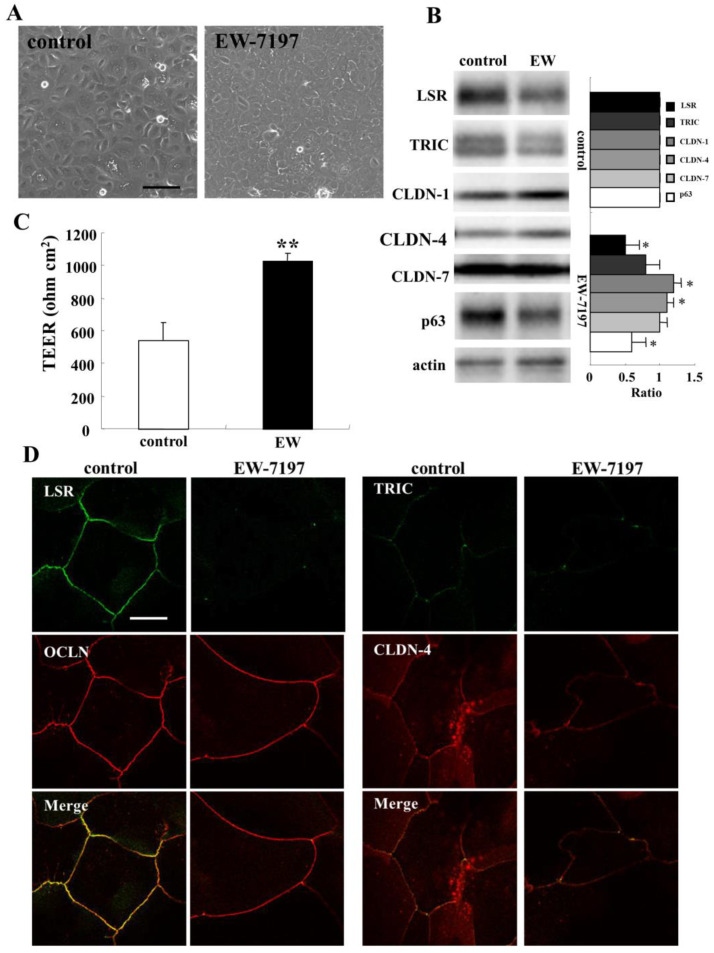
Effects of EW-7197 on tight junction molecules and epithelial barrier function in HNECs cultured with/without 10% FBS. (**A**) Phase-contrast images of HNECs (without FBS) treated with 10 µM EW-7197 for 24 h. Scale bar: 20 μm. (**B**) Western blotting for tight junction molecules and p63 in HNECs (without FBS) treated with 10 μM EW-7197 for 24 h. The corresponding expression levels are shown as bar graphs. * *p* < 0.05, ** *p* < 0.01, vs. control. (**C**) TEER values of HNECs (with FBS) treated with 10 µM EW-7197 for 24 h. ** *p* < 0.01, vs. control. (**D**) Images of immunocytochemical staining for angulin-1/LSR, OCLN, TRIC, and CLDN-4 in HNECs (with FBS) treated with 10 µM EW-7197 for 24 h. Scale bar: 20 µm.

**Figure 3 ijms-22-08390-f003:**
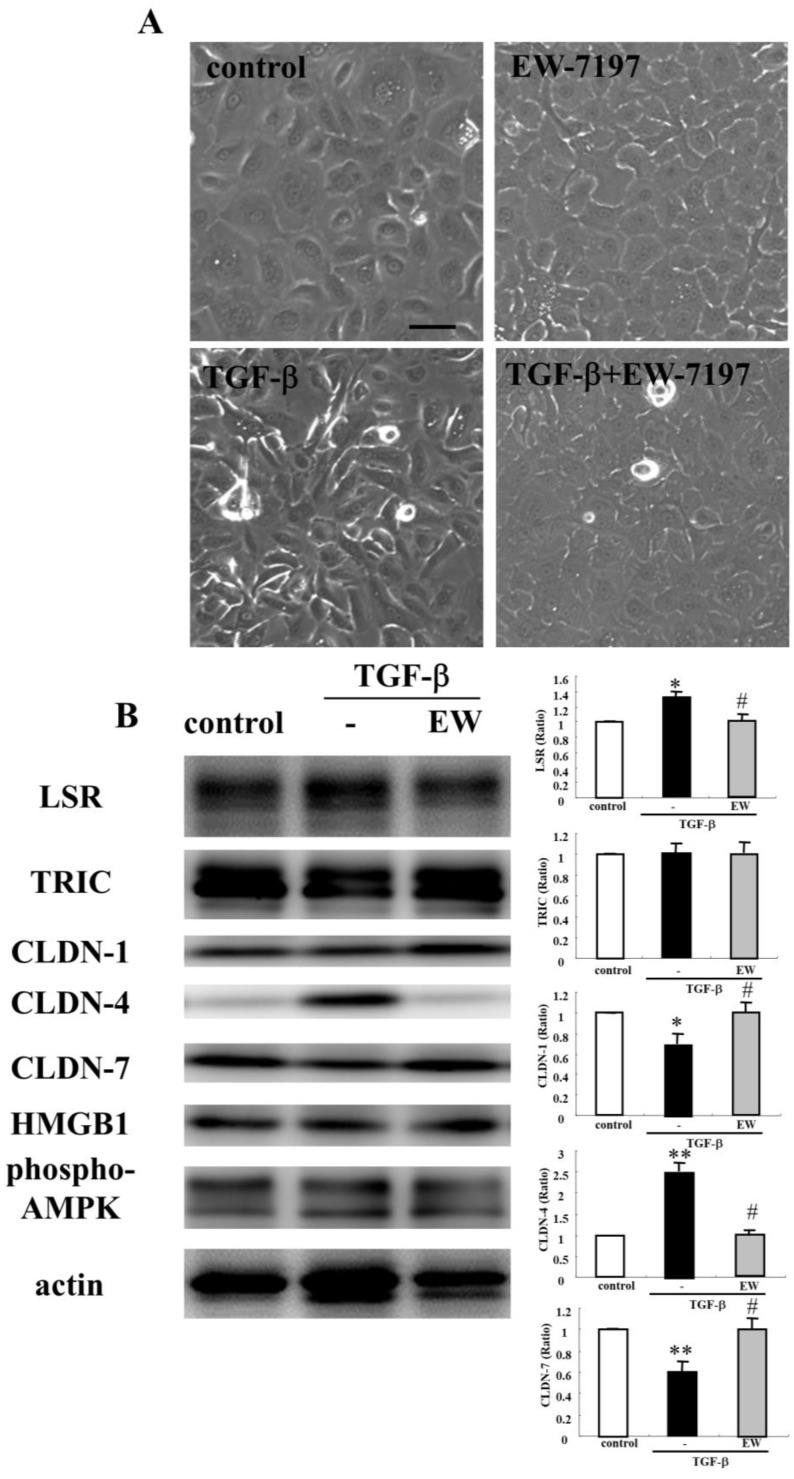
Effects of TGF-β with or without EW-7197 on tight junction molecules in HNECs cultured without 10% FBS. (**A**) Phase-contrast images of HNECs treated 100 ng/mL TGF-β1 with or without 10 μM EW-7197 for 24 h. Scale bar: 20 µm. (**B**) Western blotting for tight junction molecules, phospho-AMPK, and HMGB1 in HNECs treated 100 ng/mL TGF-β1 with or without 10 μM EW-7197 for 24 h. The corresponding expression levels are shown as bar graphs. * *p* < 0.05, ** *p* < 0.01, vs. control. # *p* < 0.05, vs. TGF-β.

**Figure 4 ijms-22-08390-f004:**
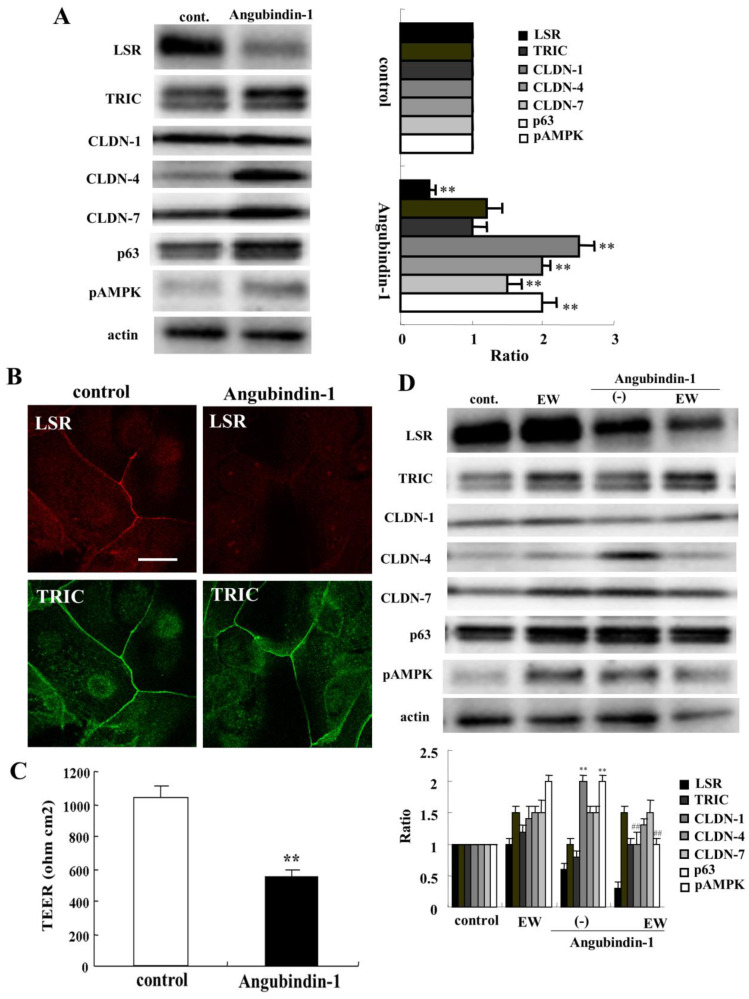
Effects of angubindin-1 with or without EW-7197 on tight junction molecules and epithelial barrier function in HNECs cultured with or without 10% FBS. (**A**) Western blotting for tight junction molecules, p63 and pAMPK in HNECs (without FBS) treated with 2.5 µg/mL angubindin-1 for 24 h. The corresponding expression levels are shown as bar graphs. ** *p* < 0.01, vs. control. (**B**) Images of immunocytochemical staining for angulin-1/LSR and TRIC in HNECs (with FBS) treated with angubindin-1. Scale bar: 20 µm. (**C**) TEER values of HNECs (with FBS) treated with 2.5 µg/mL angubindin-1 for 24 h. ** *p* < 0.01, vs. control. (**D**) Western blotting for tight junction molecules, p63 and pAMPK in HNECs (without FBS) treated with 2.5 µg/mL angubindin-1 with or without 10 μM EW-7197 for 24 h. The corresponding expression levels are shown as bar graphs. ** *p* < 0.01, vs. control. ## *p* < 0.01, vs. angubindin-1.

**Figure 5 ijms-22-08390-f005:**
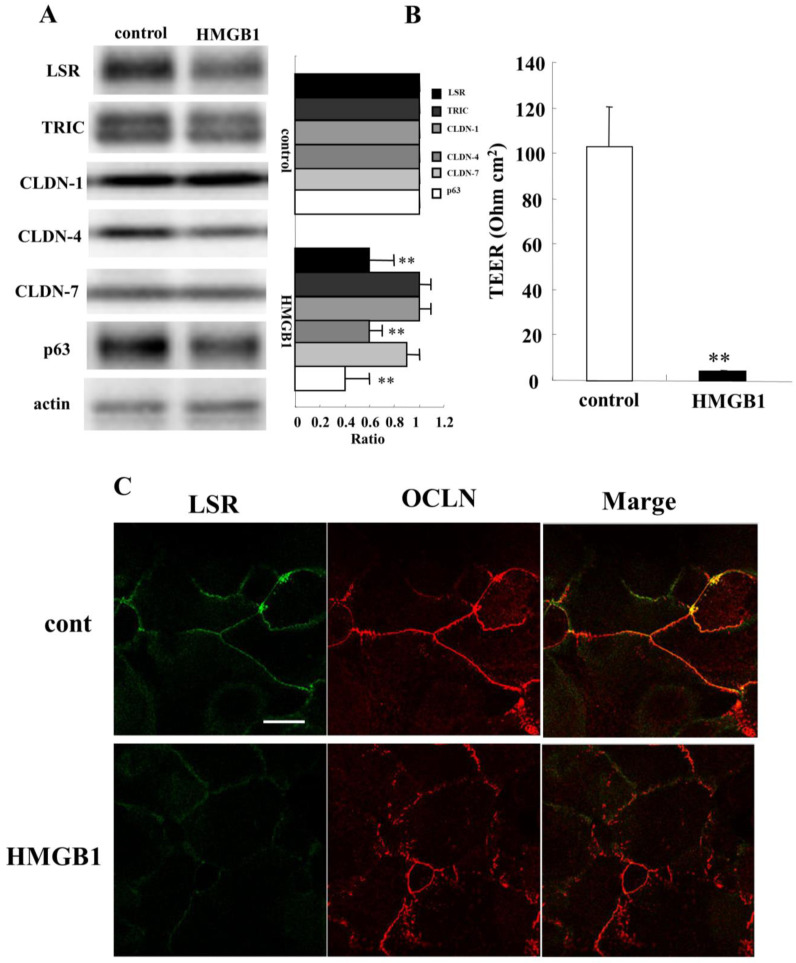
Effects of HMGB1 on tight junction molecules and epithelial barrier function in HNECs with or without 10% FBS. (**A**) Western blotting for tight junction molecules and p63 in HNECs (without FBS) treated with 100 ng/mL HMGB1 for 24 h. The corresponding expression levels are shown as bar graphs. ** *p* < 0.01, vs. control. (**B**) TEER values of HNECs (with FBS) treated with 100 ng/mL HMGB1 for 24 h. ** *p* < 0.01, vs. control. (**C**) Images of immunocytochemical staining for angulin-1/LSR and OCLN in HNECs (with FBS) treated with 100 ng/mL HMGB1 for 24 h. Scale bar: 20 µm.

**Figure 6 ijms-22-08390-f006:**
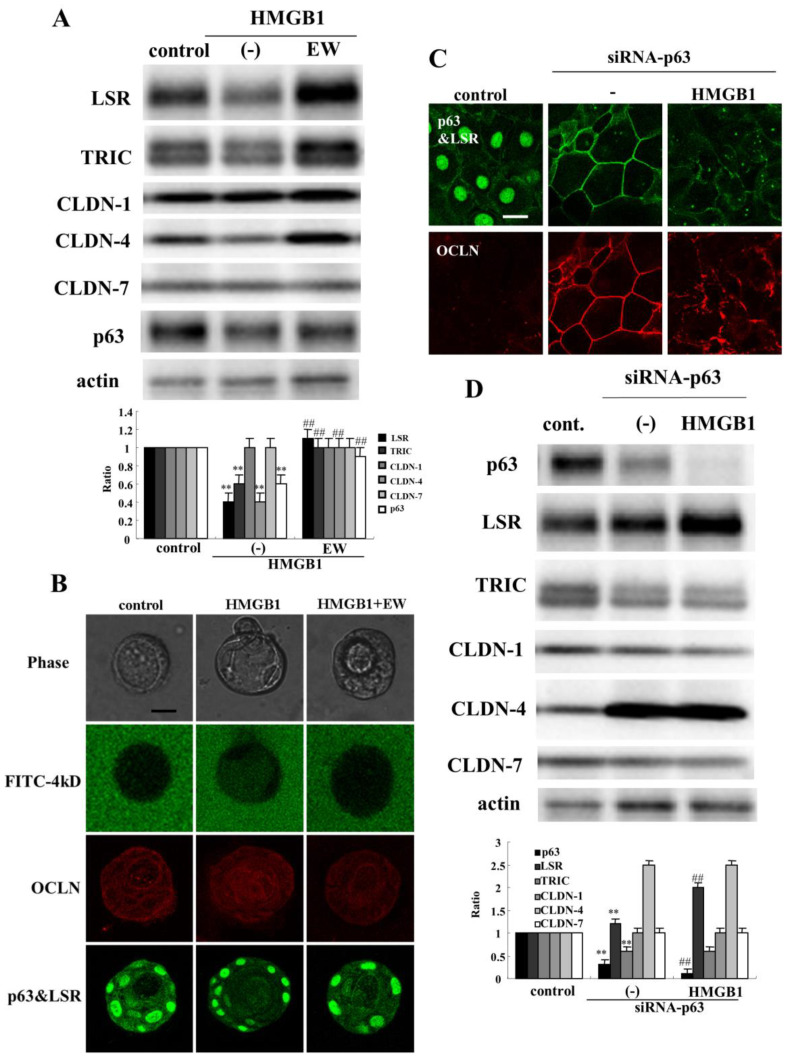
Effects of EW-7197 and siRNA-p63 on tight junction molecules and epithelial barrier function in HNECs (with or without 10% FBS) treated with HMGB1. (**A**) Western blotting for tight junction molecules and p63 in HNECs (without FBS) treated with 100 ng/mL HMGB1 with or without 10 µM EW-7197 for 24 h. The corresponding expression levels are shown as bar graphs. ** *p* < 0.01, vs. control. ## *p* < 0.01, vs. HMGB1. (**B**) Phase contrast images, FITC-dextran (4 kD) and immunocytochemical staining for OCLN, p63, angulin-1/LSR in 2.5D Matrigel culture of HNECs (without FBS) treated with 100 ng/mL HMGB1 with or without 10 µM EW-7197 for 24 h. Scale bar: 20 µm. (**C**) Images of immunocytochemical staining for p63, angulin-1/LSR and OCLN in HNECs (with FBS) transfected with siRNA-p63 with or without 100 ng/mL HMGB1for 48 h. Scale bar: 20 µm. (**D**) Western blotting for tight junction molecules and p63 in HNECs (without FBS) transfected with siRNA-p63 with or without 100 ng/mL HMGB1 for 48 h. The corresponding expression levels are shown as bar graphs. ** *p* < 0.01, vs. control. ## *p* < 0.01, vs. siRNA-p63.

**Figure 7 ijms-22-08390-f007:**
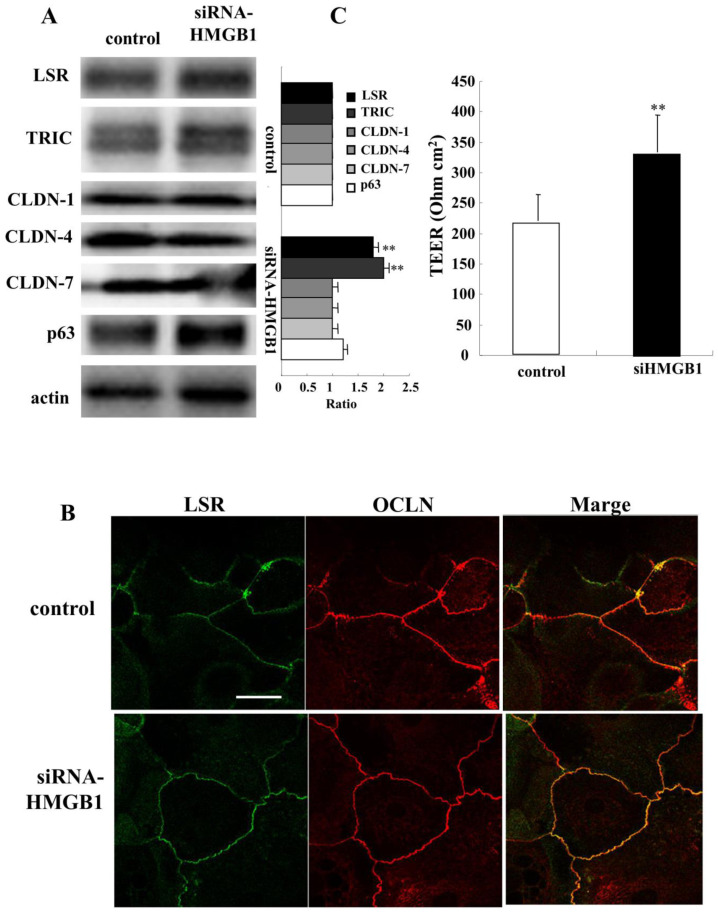
Effects of knockdown of HMGB1 on tight junction molecules and epithelial barrier function in HNECs with or without 10% FBS. (**A**) Western blotting for tight junction molecules and p63 in HNECs (without FBS) transfected with siRNA-HMGB1 for 48 h. The corresponding expression levels are shown as bar graphs. ** *p* < 0.01, vs. control. (**B**) Images of immunocytochemical staining for angulin-1/LSR and OCLN in HNECs (with FBS) transfected with siRNA-HMGB1 for 48 h. Scale bar: 20 µm. (**C**) TEER values of HNECs (with FBS) transfected with siRNA-HMGB1 for 48 h. ** *p* < 0.01, vs. control.

**Figure 8 ijms-22-08390-f008:**
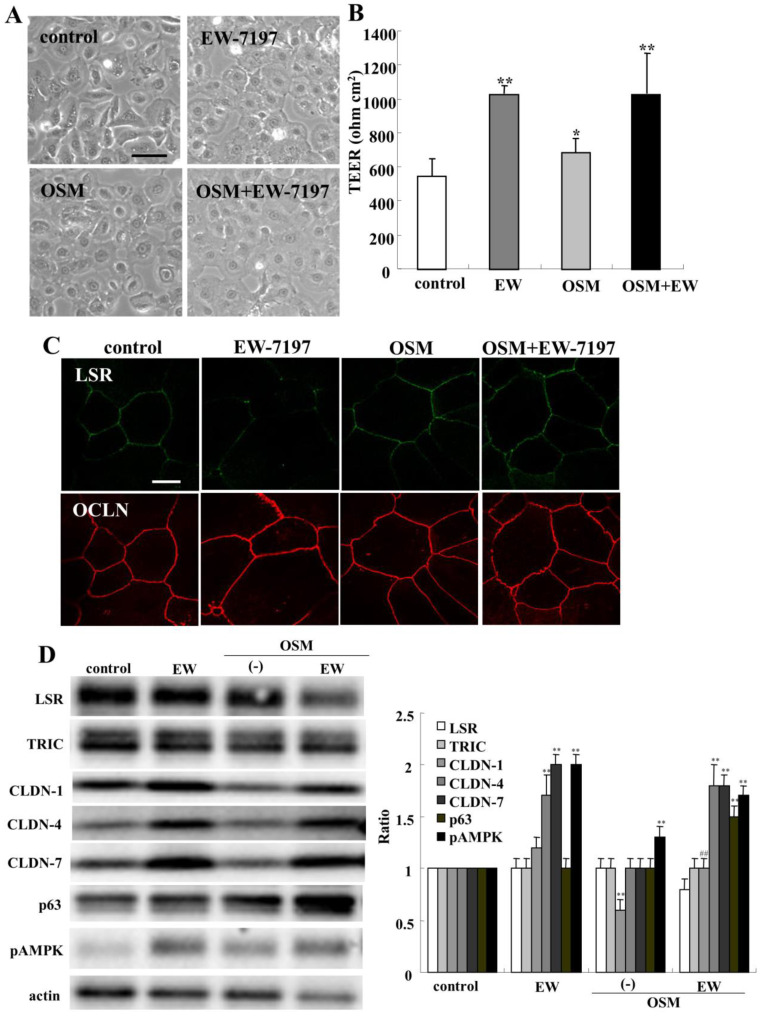
Effects of OSM with or without EW-7197 on tight junction molecules and epithelial barrier function in HNECs with or without 10% FBS. (**A**) Phase-contrast images of HNECs (without FBS) treated with 100 ng/mL OSM with or without 10 μM EW-7197. Scale bar: 20 µm. (**B**) TEER values of HNECs (with FBS) treated with 100 ng/mL OSM with or without 10 μM EW-7197. * *p* < 0.05, ** *p* < 0.01, vs. control. (**C**) Images of immunocytochemical staining for angulin-1/LSR and OCLN in HNECs (with FBS) treated with 100 ng/mL OSM with or without 10 μM EW-7197. Scale bar: 20 µm. (**D**) Western blotting for tight junction molecules, p63 and pAMPK in HNECs (without FBS) treated with 100 ng/mL OSM with or without 10 μM EW-7197. The corresponding expression levels are shown as bar graphs. ** *p* < 0.01, vs. control. ## *p* < 0.01, vs. OSM.

**Table 1 ijms-22-08390-t001:** List of gene probes in HNEC treated with HMGB1 and OSM.

Number	Gene	Control	HMGB1 + OSM	Fold (HMGB1 + OSM/Control)
NM_0218042	ACE2	3	3	1.06
NM_1536093	TMPRSS6	1	3	2.75
NM_001289823.1	FURIN	9	15	1.66
XM_011518263.1	CTSL	438	881	2.01

**Table 2 ijms-22-08390-t002:** Primers of RT-PCR.

Gene	Forward Primer	Reverse Primer
angulin-1/LSR	CAGGACCTCAGAAGCCCCTGA	AACAGCACTTGTCTGGGCAGC
tricellulin	AGGCAGCTCGGAGACATAGA	TCACAGGGTATTTTGCCACA
claudin-1	AACGCGGGGCTGCAGCTGTTG	GGATAGGGCCTTGGTGTTGGGT
claudin-4	AGCCTTCCAGGTCCTCAACT	AGCAGCGAGTCGTACACCTT
claudin-7	AGGCATAATTTTCATCGTGG	GAGTTGGACTTAGGGTAAGAGCG
G3PDH	ACCACAGTCCATGCCATCAC	TCCACCACCCTGTTGCTGTA

## Data Availability

Data is contained within the article.

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
