# Peer review of "Effects of HMGB1 on Tricellular Tight Junctions via TGF-β Signaling in Human Nasal Epithelial Cells"

_ijms, 2021, doi:10.3390/ijms22168390_

Round 1
Reviewer 1 Report
I appreciate the opportunity to review the manuscript for publication in MDPI IJMS.
The authors investigated possible relation between HMGB1 stimulation and changes in the barrier of human cultured nasal epithelial cells (HNEC) using in vitro spheroids. They also examined favorable effects of EW-7197 or siRNA-p63 treatment on epithelial
permeability barriers.
I feel that the topics is interesting and well organized.
I have a few minor comments.
Figure 1: Morphology of HNEC appears flat polygonal shape. Is this obtained from 3-D culture system?
In a series of the corresponding expression levels for immunoblotting shown as bar graphs, statistics should be clarified.
Author Response
Response to Comments and Suggestions for Authors
I appreciate the opportunity to review the manuscript for publication in MDPI IJMS.
The authors investigated possible relation between HMGB1 stimulation and changes in the barrier of human cultured nasal epithelial cells (HNEC) using in vitro spheroids. They also examined favorable effects of EW-7197 or siRNA-p63 treatment on epithelial permeability barriers. I feel that the topics is interesting and well organized.
Thank you for your interesting.
I have a few minor comments.
Figure 1: Morphology of HNEC appears flat polygonal shape. Is this obtained from 3-D culture system?
This is 2-D culture system. By a confocal laser scanning microscope, we indicated the 3-D images (Figure 1A).
In a series of the corresponding expression levels for immunoblotting shown as bar graphs, statistics should be clarified.
In all immunoblotting, we added the statistics analysis.
Reviewer 2 Report
The present study by Ohwada et al uses a nasal epithelial cell system. to investigate the importance. of LSR in TJ protein accumulation. The studies heavily rely upon a TGFb R1 inhibitor EW-7197, that induces epithelial barrier function. The authors use a interesting 2.5 D culture technique in an attempt to more closely. replicate the in vivo biology. Overall this is a straightforward. and well. performed study that uncovers one mechanism of nasal epithelil barrier regulation.
The introduction of p63 is awkward with description of p53. What is the relationship between p53 and p63, that is not clear.
The number of replicates performed and the number of times repeated and the statistical analysis is lacking thoughout the manuscript in the figure legends. The authors need to be consistent with showing the densitometric data for all of the western blotting throughout.
What is the explanation for the change in cldn 1 levels with FBS?
Fig2D, teh CLDN4 levels seem to reduce with EW, which is in. conflict with the western blotting data. This needs to be addressed.
Fig 1B needs. labelling for the z plane images
FIg3A, the shapes of teh cells are unclear and difficult to interpret.
3B, only. claudin 4 seems to be altered, all. of this data needs densitometric quantification
what is the significance of the p44 data?
Author Response
Response to Comments and Suggestions for Authors
The present study by Ohwada et al uses a nasal epithelial cell system to investigate the importance. of LSR in TJ protein accumulation. The studies heavily rely upon a TGFb R1 inhibitor EW-7197, that induces epithelial barrier function. The authors use an interesting 2.5 D culture technique in an attempt to more closely replicate the in vivo biology. Overall this is a straightforward and well performed study that uncovers one mechanism of nasal epithelial barrier regulation.
Thank you for your comments.
The introduction of p63 is awkward with description of p53. What is the relationship between p53 and p63, that is not clear.
In introduction, we added the explanation of the relationship between p53 and p63.
The number of replicates performed and the number of times repeated and the statistical analysis is lacking thoughout the manuscript in the figure legends. The authors need to be consistent with showing the densitometric data for all of the western blotting throughout.
We performed the experiments more than three times and added the statistical analysis.
What is the explanation for the change in cldn 1 levels with FBS?
In HNECs with or without FBS, CLDN-1 mRNA was detected. Among them, the change was not observed.
Fig2D, CLDN4 levels seem to reduce with EW, which is in. conflict with the western blotting data. This needs to be addressed.
In immunocytochemistry, no change of OCLN and CLDN-4 was observed at the membranes (Fig. 2D).
Fig 1B needs. labelling for the z plane images
We added the labeling for X-Y, X-Z and Y-Z.
Fig 3A, the shapes of the cells are unclear and difficult to interpret.
We deleted the sentences.
Fig 3B, only. claudin 4 seems to be altered, all. of this data needs densitometric quantification
We added the statistics analysis.
What is the significance of the p44 data?
Only pAMPK data is important. We deleted p44 data.
Round 2
Reviewer 2 Report
sufficiently improved